# Electrical Stimulation Enhances Plant Defense Response in Grapevine through Salicylic Acid-Dependent Defense Pathway

**DOI:** 10.3390/plants10071316

**Published:** 2021-06-28

**Authors:** Daisuke Mori, Ayane Moriyama, Hiroshi Kanamaru, Yoshinao Aoki, Yoshiyuki Masumura, Shunji Suzuki

**Affiliations:** 1Planning Department, Nihonshinko Co., Ltd., Osaka 590-0535, Japan; daisuke.mori@nihon-shinko.com (D.M.); hiroshi.kanamaru@nihon-shinko.com (H.K.); yoshiyuki.masumura@nihon-shinko.com (Y.M.); 2Laboratory of Fruit Genetic Engineering, The Institute of Enology and Viticulture, University of Yamanashi, Yamanashi 400-0005, Japan; g20lf007@yamanashi.ac.jp (A.M.); yaoki@yamanashi.ac.jp (Y.A.)

**Keywords:** β-1,3-glucanase, electrical stimulation, grapevine, plant defense response, salicylic acid

## Abstract

Concern over environmental pollution generated by chemical fungicides has led to the introduction of alternative pest management strategies to chemical fungicide application. One of those strategies is the induction of plant defense response by an abiotic elicitor. In the present study, field-grown grapevines were subjected to electrical stimulation using a solar panel from two weeks before flowering to harvest in the 2016 and 2020 growing seasons. In both years, electrical stimulation decreased the incidence of gray mold and/or ripe rot on bunches and downy mildew on leaves of the field-grown grapevine. Transcription of a gene encoding β-1,3-glucanase but not class IV chitinase in leaves of potted grapevine seedlings was upregulated 20 days after electrical stimulation, suggesting that electrical stimulation acts as an abiotic elicitor of plant defense response to fungal diseases. The gene expression of *PR1* but not *PDF1.2* was upregulated in *Arabidopsis* plants subjected to electrical stimulation. On the other hand, *PR1* gene expression was not induced in salicylic acid (SA)-insensitive *Arabidopsis* mutant *npr1-5* subjected to electrical stimulation. Taken together, electrical stimulation is responsible for plant defense response through the SA-dependent defense pathway. These findings would help us develop a novel and innovative practical technique that uses electrical stimulation in integrated pest management.

## 1. Introduction

The use of chemical fungicides is a simple strategy to protect grapevines from phytopathogens. However, at present, vine growers face several risks posed by the emergence of chemical fungicide-resistant phytopathogen populations. For example, *Plasmopara viticola*, which causes grape downy mildew, is a high-risk pathogen because of its high potential to acquire chemical fungicide resistance [1]. Some *P. viticola* populations in European vineyards have acquired resistance to quinine outside inhibitor (QoI) fungicide [2] and carboxylic acid amide (CAA) fungicide [3]. In Japan, QoI fungicide resistance was detected in 2009 in certain *P. viticola* populations [4]. Although CAA fungicide-resistant *P. viticola* isolates have not been reported in Japan, a single point mutation at codon 1105 of the cellulose synthase gene *PvCesA3*, which confers resistance to CAA fungicides [5], was found in heterozygotes of Japanese *P. viticola* populations [6].

Interest in eco-friendly alternatives to chemical fungicides for pest management has intensified in viticulture. One of the alternative pest management strategies is the induction of plant defense response by treatment with abiotic or biotic elicitors [7]. For example, hordenine, a phenethylamine alkaloid found in barley, suppressed grape downy mildew through the activation of plant defense response in grapevine [8]. Biological control agents also induce plant defense response in grapevine. *Trichoderma harzianum* T39 activated plant defense response in grapevine, resulting in the reduction of downy mildew severity in the grapevine without the direct inhibition of *P. viticola* [9]. Integrated pest management (IPM) has seen an upsurge of interest in viticulture. The introduction of practical techniques for inducing plant defense response in viticulture would contribute to suppressing the emergence of chemical fungicide-resistant phytopathogens by reducing chemical fungicide application.

Our objective in this study was to investigate the applicability of electrical stimulation as an abiotic elicitor in the grapevine. In our previous studies, we found that grapevine subjected to electrical stimulation using solar panels exhibited an increase in the content of resveratrol, which is one of the phytoalexins in grapevine [10], in berries compared with control grapevines [11]. Through microarray analysis, we demonstrated that electrical stimulation upregulated the transcription of genes related to stilbenoid biosynthesis in grape cells [11]. From these results, we formulated the hypothesis that electrical stimulation enhances plant defense response in grapevine as an abiotic elicitor. As far as we know, there are no studies of the effect of electrical stimulation on the incidence of fungal disease in crops. Here, we report the effect of electrical stimulation on the incidence of fungal diseases, including downy mildew in grapevine. We also demonstrate that the salicylic acid (SA)-dependent defense pathway is involved in plant defense response triggered by electrical stimulation.

## 2. Results

### 2.1. Electrical Stimulation Decreases the Incidence of Fungal Diseases in Field-Grown Grapevine

To evaluate the effect of electrical stimulation on phytopathogenic fungal diseases, field-grown grapevines were exposed to electrical stimulation from two weeks before flowering (BBCH55-57) to harvest (BBCH89) in the 2016 and 2020 growing seasons (Figure 1A). Two electrodes were screwed on one grapevine trunk (at 20 and 60 cm above ground) and connected to a solar panel. In our system, illuminance exceeding 8000 lux induced full capacity of a solar panel, whereas voltage was low when illuminance was below 8000 lux, and no voltage was detected on grapevines that had only electrodes without solar panel [11].

Disease assessment of bunches and leaves was performed at harvest (BBCH89). Electrical stimulation decreased the incidence of fungal diseases in field-grown grapevines. The incidence of gray mold and/or ripe rot on bunches in field-grown grapevines was decreased by electrical simulation in the 2016 and 2020 growing seasons compared with those in electrode-treated and control grapevines (Table 1). There was a large outbreak of grape downy mildew in the vineyard tested in 2016 growing season. In June 2016, youngberries and leaves were infected by *P. viticola*. The incidence of grape downy mildew in leaves of control grapevines was approximately 90% at harvest (Figure 1B). In contrast, the incidence of grape downy mildew in control grapevines in the 2020 growing season was approximately 25%. Irrespective of the incidence of grape downy mildew, electrical stimulation also significantly decreased the incidence of downy mildew on leaves compared with those in electrode-treated and control grapevines.

### 2.2. Induction of Plant Defense Response by Electrical Stimulation

Field-grown grapevines subjected to electrical stimulation showed enhanced accumulation of resveratrol in berries [11]. Because resveratrol is accumulated as a phytoalexin in grape berries and leaves in response to infection by phytopathogen [10], we expected that electrical stimulation would induce plant defense response in grapevines. However, total RNA could not be stably purified from the bunches and leaves of field-grown grapevines due to senescence (data not shown). Therefore, the analysis of plant defense-related gene expression was performed using potted seedlings.

Grapevines at BBCH14-15 were exposed to electrical stimulation, as shown in Figure 2A. A positive electrode was pricked on the grapevine trunk, and a negative electrode was pricked on a shoot, then connecting to a solar panel. Electrical stimulation was performed during 20 days after treatment. Total RNA was isolated from leaves of the seedlings 0, 10, and 20 days after treatment, and the transcription levels of genes encoding PR proteins, class IV chitinase and β-1,3-glucanase, were compared among the differently treated grapevines (Figure 2B). The expression of the gene encoding class IV chitinase was not upregulated by electrical stimulation during the indicated periods. In contrast, the expression of the gene encoding β-1,3-glucanase in the seedlings subjected to electrical stimulation was significantly upregulated 20 days after treatment compared with those in electrode-treated and control seedlings.

### 2.3. Electrical Stimulation Induces Plant Defense Response in Arabidopsis Plants Through SA-Dependent Defense Pathway

Typically, grapevine induces class IV chitinase and β-1,3-glucanase through the jasmonic acid (JA)-dependent defense pathway and the SA-dependent defense pathway in response to elicitors, respectively [12,13]. To determine whether electrical stimulation induces plant defense response through those pathways, wild-type *Arabidopsis* was subjected to electrical stimulation, as shown in Figure 3A. A positive electrode was pricked on the base of the inflorescence of *Arabidopsis* plants (38-day-old), and a negative electrode was pricked on the inflorescence at the distance of 15–20 cm from the positive electrode, then connecting to a solar panel. Rosette leaves of the *Arabidopsis* plants were collected 0, 12, 24, and 48 h after treatment.

Electrical stimulation upregulated the gene expression of *PR1*, a marker gene for the SA-dependent defense pathway [14], in rosette leaves of *Arabidopsis* wild-type at 12 and 48 h after treatment, compared with those of electrode-treated and control wild-type *Arabidopsis* (Figure 3B). In contrast, the transcription level of *PDF1.2*, a marker gene for the JA-dependent defense pathway [15], in rosette leaves of wild-type *Arabidopsis* subjected to electrical stimulation was comparable to those of electrode-treated and control wild-type *Arabidopsis* (Figure 3B). Electrical stimulation did not induce *PR1* gene expression in rosette leaves of SA-insensitive *npr1-5* mutant compared with non-treated mutant (Figure 3C).

Taken together, electrical stimulation is responsible for plant defense response through the SA-dependent defense pathway.

## 3. Discussion

Since field-grown grapevines have different physiological properties, including growth stages with *Arabidopsis* plants, electrical stimulation may induce different physiological changes between field-grown grapevines and *Arabidopsis* plants. Electrical stimulation increased resveratrol contents in berries of grapevines relative to those of control grapevines and electrode-treated grapevines [11], while we could not demonstrate any positive results related to physiological changes in *Arabidopsis* plants subjected to electrical stimulation. Future studies employing *Arabidopsis* pathosystem would reveal the accurate signaling pathways for plant defense response triggered by electrical stimulation in plants.

Two steel screws or needles were inserted into the trunks of grapevines or the inflorescences of *Arabidopsis* plants, wounding them. Grapevines and *Arabidopsis* plants exposed to electrical stimulation exhibited higher plant defense responses than those treated by the electrode. The result suggested that wounding by inserting steel screws or needles into plants as electrodes is not responsible for plant defense response. Therefore, electrical stimulation by solar panels might be indispensable for the induction of plant defense response in plants.

Plant defense response was not triggered by electrical stimulation in SA-insensitive *Arabidopsis* mutants. Grape cells in trunk tissue may recognize electrical stimulation through an unknown mechanism and generate SA. SA is involved in plant defense response [16]. Because SA can migrate long distances into the phloem, the transported SA may induce systemic acquired resistance (SAR) in the plant [17]. SA generated by electrical stimulation may be transported to bunches and leaves from the phloem to enhance plant defense response. Thus, electrical stimulation acts as an abiotic elicitor of plant defense response in the grapevine. In the SA-dependent defense pathway, NONEXPRESSOR OF PATHOGENESIS-RELATED GENES 1 (NPR1) is transported to the nucleus in response to SA, thereby upregulating the defense gene expression, such as PR1 and β-1,3-glucanase [18]. As β-1,3-glucanase shows direct antimicrobial activity against *B. cinerea* [19], *C. gloeosporioides* [20], and *P. viticola* [21], the inhibition of fungal diseases in bunches and leaves by electrical stimulation seems reasonable.

How does electrical stimulation activate SA biosynthesis in grapevines? So far, we do not have the answer to this question. Studies have shown that electrical signals generated in plants by mechanical damage and wounding systemically induce a broad range of plant defense responses [22,23]. However, those studies did not examine the possibility that electrical signal induces second messengers, such as SA and JA. Electrode treatment (wounding without electrical stimulation) did not decrease disease incidence in field-grown grapevines and induced plant defense response in grapevines and *Arabidopsis* plants, suggesting that electrical stimulation induced second messengers. Although future studies to detect second messengers are required, our study is the first to demonstrate crosstalk between an electrical signal and a messenger molecule in plants. Although whether or not electrical stimulation activates SA biosynthesis remains to be elucidated, the experiments on *Arabidopsis* mutant demonstrated that SA generated by electrical stimulation might be responsible for systemic plant defense response. Further investigations are necessary to elucidate whether SA generates at tissues exposed to electrical stimulation and transports into the phloem of grapevine as well as how grapevine cells recognize electrical current and how the recognition activates SA biosynthesis.

Electrical stimulation worked well to suppress several fungal diseases in the field tests. However, it is not a fast-acting tool for suppressing plant diseases. Electrical stimulation induced the expression of a gene encoding β-1,3-glucanase 20 days after treatment in potted grapevine seedlings, although we could not exclude the possibility that the intensity of the electrical stimulation applied to the plants affects the timing of expression of plant defense response. The slow action of electrical stimulation in plant disease control may be one of the problems of using electrical stimulation in the field. Under such circumstances, electrical stimulation would be suitable as a disease-preventing tool in viticulture but not as a tool for treating disease symptoms. Further investigations of combinations of electrical stimulation with common plant disease control techniques, including fungicide application in vineyards, may potentially decrease the frequency of chemical fungicide, copper, and sulfur applications and yield a new practical technique for IPM in viticulture.

In the future, eco-friendly plant disease control is expected to predominate in viticulture due to concerns over environmental pollution generated by chemical fungicides. In this study, we focused on electrical stimulation as an innovative tool for IPM in viticulture. Electrical stimulation activated the SA-dependent defense pathway and systemically suppressed fungal diseases in berries and leaves of field-grown grapevines. The voltage applied to a field-grown grapevine by electrical stimulation was oscillated by illuminance [11]. One important question that remains to be clarified is whether other environmental factors, including soil composition, weather, training system, and grapevine cultivars, affect the physiological responses related to electric stimulation. To explore further the applicability of electrical stimulation to disease control in viticulture, field tests on a number of vineyards having different environmental conditions and cultivars, adjustments of starting time and conditions for electrical stimulation, and development of a universal electrical stimulation apparatus are required.

## 4. Materials and Methods

### 4.1. Plant Materials

*Vitis vinifera* cv. Merlot was cultivated in the experimental vineyard of The Institute of Enology and Viticulture, the University of Yamanashi, Japan. The grapevines were approximately 30 years old and trained to the Guyot-style system.

Self-rooted *V. vinifera* cv. Cabernet Sauvignon seedlings were also cultivated in pots for approximately 2 months and then used for electrical stimulation.

Seeds of wild-type *Arabidopsis thaliana* (Col-0) and SA-insensitive mutant *npr1-5* (CS3724) [14] were obtained from The Arabidopsis Information Resource (TAIR), sown on rockwool blocks, and then incubated at 22 °C in an incubator (11.8 Wm^−2^ for 16 h in a day). The seedlings were planted in soil, and 38-day-old plants were used for electrical stimulation.

### 4.2. Electrical Stimulation of Field-Grown Grapevines and Grapevine Seedlings

Six grapevines were prepared for electrical stimulation. Electrical stimulation was carried out on 20 May 2016 and 12 May 2020 (approximately two weeks before flowering; BBCH55-57) according to a previously described method with slight modification [11]. Briefly, two electrodes (steel screws, 40 mm length, 3.3 mm diameter) were entirely screwed on one grapevine trunk (at 20 and 60 cm above ground) and connected to a solar panel (upper, negative electrode; lower, positive electrode; Figure 1A). The solar panels were located 2.5 m above ground. The solar panel had the following electrical characteristics: maximum voltage 11.6 V ± 5%, maximum current 100 mA ± 5%, and working temperature −35 °C to 85 °C. Grapevines with only electrodes (without solar panel) or without any treatment were prepared as controls. Each grapevine received the same treatment in both years. Electrical stimulation was performed from BBCH55-57 to BBCH89 in both years.

Three grapevine seedlings with 4–5 expanding leaves at BBCH14-15 were subjected to electrical stimulation. A positive electrode (steel nail, 4 mm length, 1 mm diameter) was entirely pricked on the grapevine trunk, and a negative electrode (steel needle, 2 mm length, 0.4 mm diameter) was entirely pricked on a shoot (Figure 2A). The electrodes were connected to a solar panel. The seedlings were cultivated at 27 °C for 10 d and 20 d in an incubator (11.8 Wm^−2^ for 16 h in a day). Seedlings with only electrodes (without solar panel) or without any treatment were prepared as controls. The third to the fifth leaves of the grapevines were detached and frozen immediately in liquid nitrogen for real-time RT-PCR. Electrical stimulation was performed 20 days after the treatment. Three independent experiments were performed.

### 4.3. Electrical Stimulation of Arabidopsis Plants

Three 38-day-old *Arabidopsis* plants were subjected to electrical stimulation. A positive electrode (steel needle, 1 mm length, 0.4 mm diameter) was entirely pricked on the base of the inflorescence, and a negative electrode (steel needle, 1 mm length, 0.4 mm diameter) was entirely pricked on the inflorescence at the distance of 15–20 cm from the positive electrode (Figure 3A). The electrodes were connected to a solar panel. The *Arabidopsis* plants were cultivated at 22 °C for 12 h, 24 h, and 48 h in an incubator (11.8 Wm^−2^ for 16 h in a day). *Arabidopsis* plants with only electrodes (without solar panel) or without any treatment were prepared as controls. Electrical stimulation was performed 48 h after treatment. Rosette leaves of the *Arabidopsis* plants were detached and frozen immediately in liquid nitrogen for real-time RT-PCR. Three independent experiments were performed.

### 4.4. Disease Assessment

Disease assessment of bunches and leaves was conducted at harvest (BBCH89) on 9 September 2016 and 17 September 2020, respectively. All bunches were collected, and the number of bunches infected with *Botrytis cinerea* and/or *Colletotrichum*
*gloeosporioides* was counted manually. All leaves on each grapevine were assessed for downy mildew. The number of leaves infected with *P. viticola* was counted manually. Disease incidences were calculated using the following formula:

Incidence (%) = number of infected bunches or leaves/total number of bunches or leaves on one grapevine × 100

### 4.5. Real-Time RT-PCR

Total RNA was extracted from leaves of grapevine and *Arabidopsis* with a Fruit-mate for RNA Purification (Takara, Otsu, Japan), and this was followed by isolation and purification on a NucleoSpin RNA Plant (Takara) according to the manufacturer’s instructions. First-strand cDNA was synthesized from total RNA using a PrimeScript RT Reagent Kit with gDNA Eraser (Takara). Real-time RT-PCR was performed with SYBR Premix Ex Taq II (Takara). PCR amplification was performed for 40 cycles at 95 °C for 5 s and at 60 °C for 1 min after an initial denaturation at 95 °C for 30 s. The nucleotide sequences of the primers used for real-time RT-PCR were as follows: *V. vinifera* class IV chitinase primers (5′-CAATCGGGTCCTTGTGATTC-3′ and 5′-CAAGGCACTGAGAAACGCT-3′, GenBank accession no. U97522); *V. vinifera* β-1,3-glucanase primers (5′-GAATCTGTTCGATGCCATGC-3′ and 5′-GCATTATCAACCGTAGTCCC-3′, GenBank accession no. DQ267748); *V. vinifera* β-actin primers (5′-CAAGAGCTGGAAACTGCAAAGA-3′ and 5′-AATGAGAGATGGCTGGAAGAGG-3′, GenBank accession no. AF369524); *A. thaliana* PR1 primers (5′-CCTGGGGTAGCGGTGACTT-3′ and 5′-CGTGTTCGCAGCGTAGTTGT-3′, GenBank accession no. NM_127025); *A. thaliana* PDF1.2 primers (5′-TCACCCTTATCTTCGCTGCTC-3′ and 5′-ACCATGTCCCACTTGGCTTC-3′, GenBank accession no. AY063779); and *A. thaliana* actin primers (5′-GCCGACAGAATGAGCAAAGAG-3′ and 5′-AGGTACTGAGGGAGGCCAAGA-3′, GenBank accession no. NM_179953). Data were analyzed using Thermal Cycler Dice RealTime System Single Software ver. 3.00 (Takara) according to the manufacturer’s instructions. Each actin was used for data normalization. The dissociation curves for each sample were evaluated to verify the specificity of the amplification reaction. Using the standard curve method, the expression levels of each gene were determined as the number of amplification cycles needed to reach a fixed threshold. Relative gene expression was expressed as relative values to actin expression values at each sampling time.

### 4.6. Statistical Analysis

Data are presented as means ± standard deviations. Statistical analysis was performed by using Excel statistics software 2012 (Social Survey Research Information, Tokyo, Japan). Disease incidence on bunches was subjected to the chi-square test. Disease incidence on leaves and the expression levels of genes tested were subjected to the parametric Tukey’s multiple comparison test.

## Figures and Tables

**Figure 1 plants-10-01316-f001:**
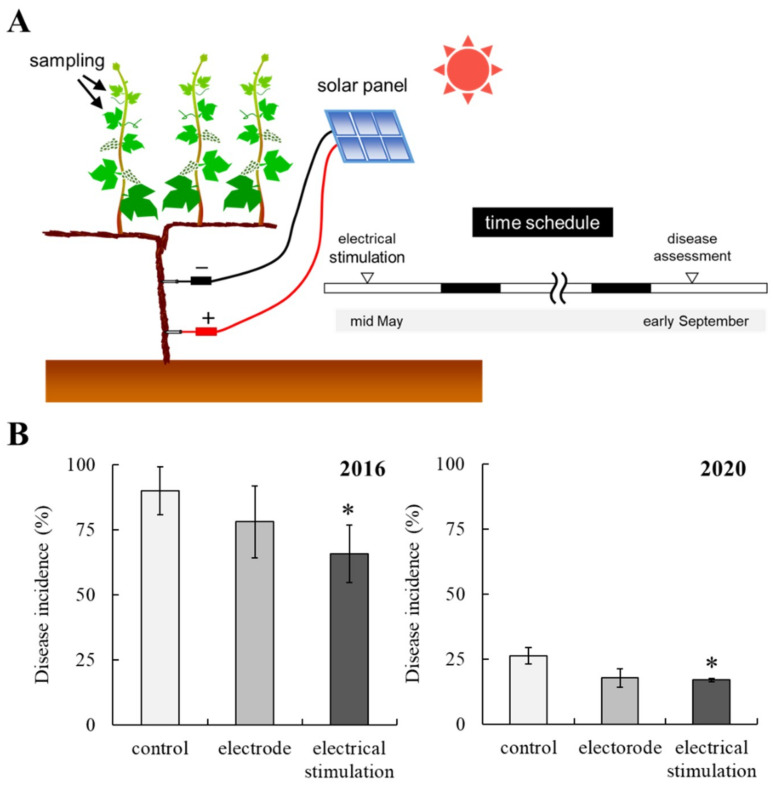
The decrease in fungal disease incidence by downy mildew in field-grown grapevine by electrical stimulation. (**A**) A schematic representation of electrical stimulation applied to field-grown grapevines. The time schedule for the experiment is also shown. (−, negative electrode. +, positive electrode.) (**B**) Incidence of downy mildew on leaves of field-grown grapevines subjected to electrical stimulation in the 2016 and 2020 growing seasons. The number of leaves infected with *P. viticola* was counted, and disease incidence was calculated as described in Materials and Methods. Bars indicate means ± standard deviations. The mean value statistically different from control is indicated by an asterisk (*p* < 0.05). Control, non-treated (without stimulation). Electrode, electrode-treated without solar panel (wounding without electrical stimulation). Electrical stimulation, electrode + solar panel (wounding + electrical stimulation).

**Figure 2 plants-10-01316-f002:**
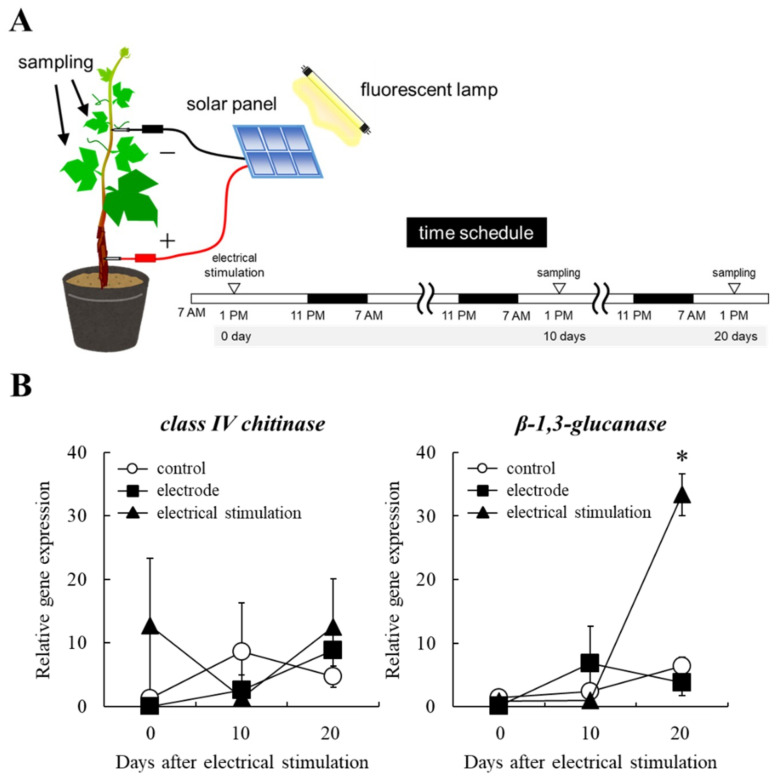
The transcription of genes encoding PR proteins in leaves of potted grapevine seedlings subjected to electrical stimulation. (**A**) A schematic representation of electrical stimulation applied to potted grapevine seedlings. The time schedule for the experiment is also shown. (−, negative electrode. +, positive electrode.) (**B**) The transcription of genes encoding class IV chitinase and β-1,3-glucanase in leaves of seedlings. Real-time RT-PCR was performed using leaves of grapevines subjected to electrical stimulation as described in Materials and Methods. Bars indicate means ± standard deviations calculated for three independent experiments. The mean value statistically different from control is indicated by an asterisk (*p* < 0.05). Control, non-treated (without stimulation). Electrode, electrode-treated without solar panel (wounding without electrical stimulation). Electrical stimulation, electrode + solar panel (wounding + electrical stimulation).

**Figure 3 plants-10-01316-f003:**
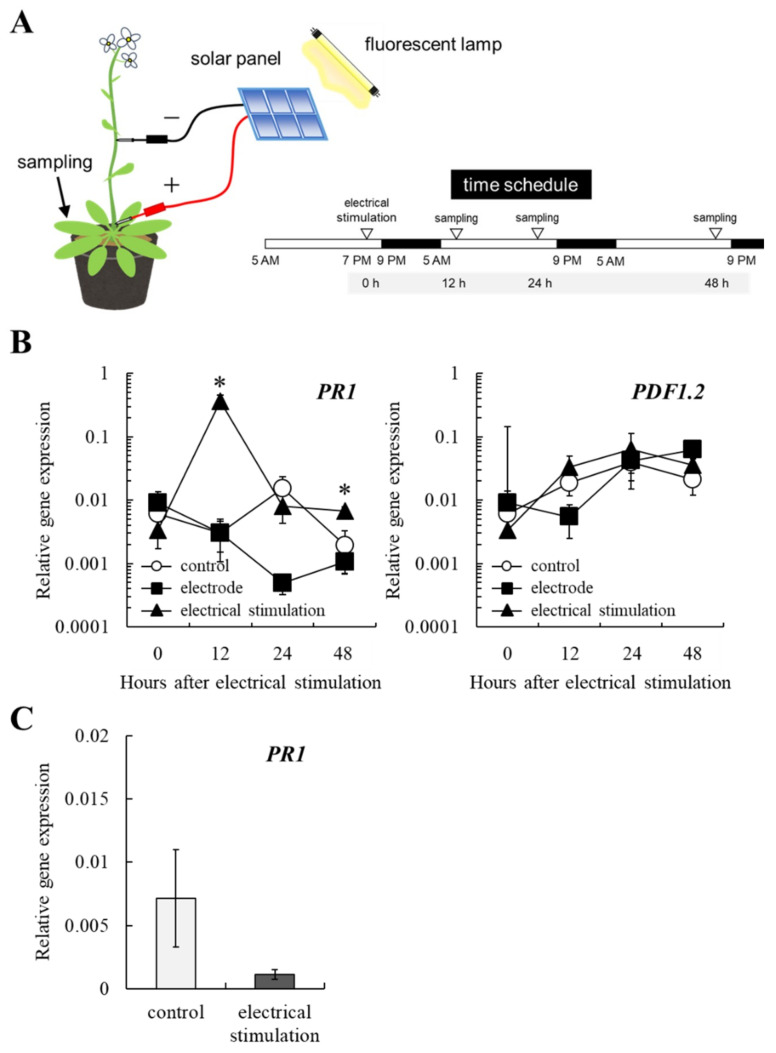
Electrical stimulation is responsible for plant defense response through the SA-dependent defense pathway. (**A**) A schematic representation of electrical stimulation applied on *Arabidopsis* plants. The time schedule for the experiment is also shown. (−, negative electrode. +, positive electrode.) (**B**) The transcription of *PR1* and *PDF1.2* genes in rosette leaves of wild-type *Arabidopsis*. Real-time RT-PCR was performed using rosette leaves of wild-type *Arabidopsis* subjected to electrical stimulation as described in Materials and Methods. Bars indicate means ± standard deviations calculated for three independent experiments. Mean values statistically different from control are indicated by an asterisk (*p* < 0.05). Control, non-treated. Electrode, electrode-treated. Electrical stimulation, electrode + solar panel. (**C**) The transcription of *PR1* gene in rosette leaves of *npr1-5* mutant. Real-time RT-PCR was performed using rosette leaves of *npr1-5* mutant subjected to electrical stimulation for 12 h as described in Materials and methods. Bars indicate means ± standard deviations calculated for three independent experiments. Control, non-treated (without stimulation). Electrical stimulation, electrode + solar panel (wounding + electrical stimulation).

**Table 1 plants-10-01316-t001:** The effect of electrical stimulation on disease incidence of bunches gray mold and/or ripe rot in field-grown grapevines.

Year	Treatment	Number of Infected Bunch	Number of Healthy Bunch	Incidence (%)
2016	control	23	10	69.7
	electrode	18	9	80.9
	electrical stimulation	22	27	44.9 *
2020	control	47	5	90.4
	electrode	28	12	76.0
	electrical stimulation	33	16	67.3 *

* indicates a significant difference from control and electrode according to chi-square test.

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
