# Peer review of "Electrical Stimulation Enhances Plant Defense Response in Grapevine through Salicylic Acid-Dependent Defense Pathway"

_plants, 2021, doi:10.3390/plants10071316_

Round 1

Reviewer 1 Report

The manuscript entitled "Electrical stimulation enhances plant defense response in 2 grapevine through salicylic acid-dependent defense pathway" report some new and very interesting data about the effects of electric fileds in inducing disease resistance mechanisms against fungal phytopathogens. The topic is new and very interesting, suggesting new area of research in the field of the Plant Pathology.  However, the manuscript could have been  prepared better than it was done. One of the major constrainst is the lack of details and information about the methods used to quantify the expression level of the genes involved in the resistance mechanisms. Moreover, information about the experimental design and the number of biological and technical replicates are missing and the inference presented in the discussuion section need to be reconsidered, as no data are presented about the mobilitation of the SA pathways in the observed reduction of the diseases. Many other point are reported in the annotated manuscript.

Author Response

Point-by-point response to the comments (Reviewer #1):

Reviewers' comments:

  1. The manuscript entitled "Electrical stimulation enhances plant defense response in grapevine through salicylic acid-dependent defense pathway" report some new and very interesting data about the effects of electric fileds in inducing disease resistance mechanisms against fungal phytopathogens. The topic is new and very interesting, suggesting new area of research in the field of the Plant Pathology. However, the manuscript could have been prepared better than it was done. One of the major constrainst is the lack of details and information about the methods used to quantify the expression level of the genes involved in the resistance mechanisms. Moreover, information about the experimental design and the number of biological and technical replicates are missing and the inference presented in the discussuion section need to be reconsidered, as no data are presented about the mobilitation of the SA pathways in the observed reduction of the diseases. Many other point are reported in the annotated manuscript.

Answer: Thank you very much for your advice to improve our manuscript. We revised the Materials and methods and Results sections for the readers to understand our study easily.

        So far, we could not demonstrate any positive results related to SA generation and transport in grapevine by our technical issues. Therefore, we added the description about it as follows:

‘Further investigations are necessary to elucidate whether SA generates at tissues exposed to electrical stimulation and transports into phloem of grapevine as well as how grapevine cells recognize electrical current and how the recognition activates SA biosynthesis.’

(See p. 9, lines 217-220, please)

The following is our point-by-point response to the comments and detailing all changes made on the revised manuscript.

Results:

Figure 3. Arabidopsis plants induced more transcripts (relative values to actin.) of plant defense-related genes in response to electrical stimulation than grapevine seedlings. Since normal scales in Y-axes hid slight changes of gene expression, we used logarithmic scale in Y-axes in Figures 3B and 3C.

Also, we changed ‘Relative intensity’ to ‘Relative gene expression’ in the revised Figures 2 and 3.

        (See the revised Figure 2 and Figure 3, please)

Discussion:

According to you suggestion, we added some discussion about points of difference between field-grown grapevines and Arabidopsis plants in the Discussion section as follows: 

‘Since field-grown grapevines have different physiological properties including growth stages with Arabidopsis plants, electrical stimulation may induce different physiological changes between field-grown grapevines and Arabidopsis plants. Electrical stimulation increased resveratrol contents in berries of grapevines relative to those of control grapevines and electrode-treated grapevines [11], while we could not demonstrate any positive results related to physiological changes in Arabidopsis plants subjected to electrical stimulation. Future studies employing Arabidopsis pathosystem would reveal the accurate signaling pathways for plant defense response triggered by electrical stimulation in plants.’

(See p. 8, lines 172-180, please)

Also, we revised the Discussion section all over.

(See the revised Discussion section, please)

So far, we could not demonstrate any positive results related to SA generation and transport in grapevine by our technical issues. Therefore, we added the description about it as follows:

‘Further investigations are necessary to elucidate whether SA generates at tissues exposed to electrical stimulation and transports into phloem of grapevine as well as how grapevine cells recognize electrical current and how the recognition activates SA biosynthesis.’

(See p. 9, lines 217-220, please)

Also, we revised the paragraph according to you suggestion as follows:

‘How does electrical stimulation activate SA biosynthesis in grapevine? So far, we do not have the answer to this question. Studies have shown that electrical signals generated in plants by mechanical damage and wounding systemically induce a broad range of plant defense responses [22,23]. However, those studies did not examine the possibility that electrical signal induces second messengers, such as SA and JA. Electrode treatment (wounding without electrical stimulation) didn’t decrease disease incidence in field-grown grapevines and induce plant defence response in grapevines and Arabidopsis plants, suggesting that electrical stimulation induced second messengers. Although future studies to detect second messengers are required, our study is the first to demonstrate crosstalk between an electrical signal and a messenger molecule in plants. Although whether or not electrical stimulation activates SA biosynthesis remains to be elucidated, the experiments on Arabidopsis mutant demonstrated that SA generated by electrical stimulation may be responsible for systemic plant defense response. Further investigations are necessary to elucidate whether SA generates at tissues exposed to electrical stimulation and transports into phloem of grapevine as well as how grapevine cells recognize electrical current and how the recognition activates SA biosynthesis.’

(See p. 8-9, lines 205-220, please)

We have taken your suggestion, and added the new sentence as follows:

‘Electrical stimulation induced the expression of a gene encoding β-1,3-glucanase 20 days after treatment in potted grapevine seedlings, although we could not exclude the possibility that the intensity of the electrical stimulation applied to the plants affects the timing of expression of plant defense response.’

(See p. 9, lines 222-226, please)

Materials and methods:

The first reason is that Merlot showed higher incidences of grey mold, ripe rot, and downy mildew diseases than Cabernet Sauvignon in our experimental vineyard every years. The second reason is that Merlot seedlings were not timely prepared for electrical stimulation.

         We guess no difference on the effect of electrical stimulation among V. vinifera cultivars.

We included details about the duration of the electrical stimulation in the revised Material and methods. For example, we added the duration of electrical stimulation to field-grown grapevines as follows:

‘Electrical stimulation was performed from BBCH55-57 to BBCH89 in both years.’

                 (See p. 10, lines 281-282, please)

Now, procedure of real-time PCR has been conventional. Therefore, the description about real-time PCR data analysis was revised as follows:

‘Data were analyzed using Thermal Cycler Dice Real Time System Single Software ver. 3.00 (Takara) according to the manufacturer’s instructions. Each actin was used for data normalization. The dissociation curves for each sample were evaluated to verify the specificity of the amplification reaction. Using the standard curve method, the expression levels of each gene were determined as the number of amplification cycles needed to reach a fixed threshold. Data were expressed as relative values to actin.’

(See p. 11, lines 328-335, please)

Details about the number of biological and technical replicates used in the experiments were included in each sub-section of the Material and methods section.

(See the revised Material and methods section, please)

    Tukey’s test is a multiple comparison test and show multiple comparison among samples. We obtained multiple comparison (control vs. electrode, control vs. electrical stimulation from three samples, and electrode vs. electrical stimulation) in Figure 3B. However, the homogenous group with the appropriate letters was difficult for us to understand which has significant difference among the samples. Therefore, we indicated mean values statistically different from control by an asterisk in Figures 1-3 .  

  1.  peer-review-12177165.v1.pdf

Answer: The manuscript was revised according to your suggestions. We highlighted the changes to the revised manuscript within the document by using the track changes mode in MS Word.

We hope that our revision is satisfactory and the revised manuscript is now acceptable for publication in the Plants.

Reviewer 2 Report

The manuscript “Electrical stimulation enhances plant defense response in grapevine through salicylic acid-dependent defense pathway” of Mori et all describes  how a novel method could contribute to disease control by enhancing plant defence in grapevine as an alternative to chemical treatments. The  manuscript is generally well written and I have only a few remarks for improvement.

Partly due to the structure of the manuscript with material and methods at the end, some details from the experiments will need clarification and description in further detail. For example the differences between the treatments control, electrode and electrical stimulation only become clear  in the material and methods section at the end of the manuscript although it would be necessary to mention this in more explicative captions and in the text, at least when first mentioned. The schedules for electrical stimulation especially in the field are not well describes and remain unclear. Further it is not stated if it was the same or different plants were used in the field experiment in the years 2016 and 2020.

The rather strong effects simply from placing the electrode should be discussed. just as the differences between the experiments from the two different years.

Although the steps from field experiment to laboratory and from grapevine to Arabidopsis mutants are logic and not unusual, the reasoning, relevance and the limitations by this setup need to be discussed in more detail since different effects, different time spans between stimulation and effect-measurement and different plants/plant parts at different growth stages are compared.

Apart from this the findings are very interesting and the understanding of the mechanisms is a prerequisite for improving the effectivity of alternatives in disease control. They could therefore contribute to necessary improvements in sustainable integrated pest management.

Detailed remarks and corrections:

L58-9 Please cite if there has been shown any effects in other crops (eg tomato).

L71-3 please give more precise indication where (eg whole plant, gapes, berries) and when (crop stage) the incidence was compared

L 82-90 It should be explained in the caption what control and electrode actually mean (similar wounding without electrical simulation). 

L132 “conclude is not the right wording for the “Results” part. Indicate eg could be more appropriate

L174-6  It would be nice if you could give more precise indications on what kind of studies would be necessary

L 187-8 This is very general, especially since organically produced wine already has quite a large market share. A reference to widely used copper and sulphur and some persistence issues would be good and improve the relevance.

L197-200 Why were different cv used in field and pot. Which effect could this have had on the results?

L202-6 please precise crop stages (BBCH, EC), not just age!

L208-17 how long was the stimulation period? A day one hour? was it eh same vines in 2016 and 2020 , did the electrodes stay all the time. This is important for me to better understand the experiment and important information needs to be included and should not be looked up from references!

Author Response

Point-by-point response to the comments (Reviewer #2):

Reviewers' comments:

General:

  1. The manuscript “Electrical stimulation enhances plant defense response in grapevine through salicylic acid-dependent defense pathway” of Mori et all describes how a novel method could contribute to disease control by enhancing plant defence in grapevine as an alternative to chemical treatments. The manuscript is generally well written and I have only a few remarks for improvement.

Answer: Thank you very much for your advice to improve our manuscript. The manuscript was revised according to the reviewers’ suggestions. We hope that our revision is satisfactory and the revised manuscript is now acceptable for publication in the Plants.

  1. Partly due to the structure of the manuscript with material and methods at the end, some details from the experiments will need clarification and description in further detail. For example the differences between the treatments control, electrode and electrical stimulation only become clear in the material and methods section at the end of the manuscript although it would be necessary to mention this in more explicative captions and in the text, at least when first mentioned. The schedules for electrical stimulation especially in the field are not well describes and remain unclear. Further it is not stated if it was the same or different plants were used in the field experiment in the years 2016 and 2020.

Answer: We agree with your suggestion, and revised Results section for the readers to understand the Methods easily.

(See p. 2, lines 66-73, p. 4, lines 110-113, and p. 6, lines 140-144, please)

         The information of grapevines was included in the revised Materials and methods section as follows:

         ‘Each grapevine received the same treatment in both years.’

(See p. 10, lines 269-270, please)

  1. The rather strong effects simply from placing the electrode should be discussed. just as the differences between the experiments from the two different years.

Answer: We have taken your suggestion, and added the discussion about differential effect between electrical stimulation and electrode-treatment in the Discussion section as follows:

 ‘Two steel screws or needles were inserted into the trunks of grapevines or the inflorescences of Arabidopsis plants, wounding them. Grapevines and Arabidopsis plants exposed to electrical stimulation exhibited higher plant defense response than those treated by electrode. The result suggested that wounding by inserting steel screws or needles into plants as electrodes are not responsible for plant defense response. Therefore, electrical stimulation by solar panel might be indispensable for induction of plant defense response in plants.’

(See p. 8, lines 181-187, please)

  1. Although the steps from field experiment to laboratory and from grapevine to Arabidopsis mutants are logic and not unusual, the reasoning, relevance and the limitations by this setup need to be discussed in more detail since different effects, different time spans between stimulation and effect-measurement and different plants/plant parts at different growth stages are compared.

Answer: According to you suggestion, we added some discussion about points of difference between field-grown grapevines and Arabidopsis plants in the Discussion section as follows: 

‘Since field-grown grapevines have different physiological properties including growth stages with Arabidopsis plants, electrical stimulation may induce different physiological changes between field-grown grapevines and Arabidopsis plants. Electrical stimulation increased resveratrol contents in berries of grapevines relative to those of control grapevines and electrode-treated grapevines [11], while we could not demonstrate any positive results related to physiological changes in Arabidopsis plants subjected to electrical stimulation. Future studies employing Arabidopsis pathosystem would reveal the accurate signaling pathways for plant defense response triggered by electrical stimulation in plants.’

        (See p. 8, lines 172-180, please)

  1. Apart from this the findings are very interesting and the understanding of the mechanisms is a prerequisite for improving the effectivity of alternatives in disease control. They could therefore contribute to necessary improvements in sustainable integrated pest management.

Answer: According to you suggestion, we added some discussion about future studies in the Discussion section as follows:

‘In the future, eco-friendly plant disease control is expected to predominate in viticulture due to concerns over environmental pollution generated by chemical fungicides. In this study, we focused on electrical stimulation as an innovative tool for IPM in viticulture. Electrical stimulation activated the SA-dependent defense pathway and systemically suppressed fungal diseases in berries and leaves of field-grown grapevines. Voltage applied to a field-grown grapevine by electrical stimulation was oscillated by illuminance [11]. One important question that remains to be clarified is whether other environmental factors including soil composition, weather, training system, and grapevine cultivars affect the physiological responses related to the electric stimulation. To explore further the applicability of electrical stimulation to disease control in viticulture, field tests on a number of vineyards having different environmental conditions and cultivars, adjustments of starting time and conditions for electrical stimulation, and development of a universal electrical stimulation apparatus are required.’ 

(See p. 9, lines 234-246, please)

Detailed remarks and corrections:

  1. L58-9 Please cite if there has been shown any effects in other crops (eg tomato).

Answer: We don’t know any studies of the effect of electrical stimulation on the incidence of fungal disease in crops. We revised the sentence as follows:

‘As far as we know, there are no studies of the effect of electrical stimulation on the incidence of fungal disease in crops.’

(See p. 2, lines 58-60, please)

  1. L71-3 please give more precise indication where (eg whole plant, gapes, berries) and when (crop stage) the incidence was compared

Answer: We revised the sentence as follows:

       ‘There was a large outbreak of grape downy mildew in the vineyard tested in 2016 growing season. In June 2016, young berries and leaves were infected by P. viticola. The incidence of grape downy mildew in leaves of control grapevines was approximately 90% at harvest (Figure 1B).’

(See p. 2, lines 79-82, please)

  1. L 82-90 It should be explained in the caption what control and electrode actually mean (similar wounding without electrical simulation).

Answer: We have taken the reviewer’s suggestion, and revised them as follows:

‘Control, non-treated(without stimulation). Electrode, electrode-treated without solar panel (wounding without electrical stimulation). Electrical stimulation, electrode + solar panel (wounding + electrical stimulation).’

Then, all captions of Figures were revised in the same manner.

 (See Figure captions, please)

  1. L132 “conclude is not the right wording for the “Results” part. Indicate eg could be more appropriate.

Answer: We removed ‘conclude’ from the revised manuscript as follows:

Taken together, electrical stimulation is responsible for plant defense response through the SA-dependent defense pathway.

(See p. 7, lines 153-154, please)

  1. L174-6 It would be nice if you could give more precise indications on what kind of studies would be necessary

Answer: Thank you for your suggestion. We revised the sentence as follows:

        ‘Further investigations are necessary to elucidate whether SA generates at tissues exposed to electrical stimulation and transports into phloem as well as how plant cells recognize electrical current and how the recognition activates SA biosynthesis.’

(See p. 9, lines 217-220, please)

  1. L 187-8 This is very general, especially since organically produced wine already has quite a large market share. A reference to widely used copper and sulphur and some persistence issues would be good and improve the relevance.

Answer: According to the reviewer’s suggestion, we revised the sentence as follows:

       ‘Further investigations of combinations of electrical stimulation with common plant disease control techniques including fungicide application in vineyards may potentially decrease the frequency of chemical fungicide, copper, and sulphur applications and yield a new practical technique for IPM in viticulture.’

       (See p. 9, lines 229-233, please)

  1. L197-200 Why were different cv used in field and pot. Which effect could this have had on the results?

Answer: The first reason is that Merlot showed higher incidences of grey mold, ripe rot, and downy mildew diseases than Cabernet Sauvignon in our experimental vineyard every years. The second reason is that Merlot seedlings were not timely prepared for electrical stimulation.

       We guess no difference on the effect of electrical stimulation among V. vinifera cultivars.

  1. L202-6 please precise crop stages (BBCH, EC), not just age!

Answer: Thank you for your suggestion. In the revised manuscript, we included crop stages (BBCH-scale for grapevine).

  1. L208-17 how long was the stimulation period? A day one hour? was it eh same vines in 2016 and 2020 , did the electrodes stay all the time. This is important for me to better understand the experiment and important information needs to be included and should not be looked up from references!

Answer: Periods of electrical stimulation for field-grown grapevines, grapevine seedlings, and Arabidopsis plants were included in the revised manuscript, respectively.

(See p. 9, lines 270-271, 281-282, and 291-292, please)

Reviewer 3 Report

This article reports about the responses to the electrical stimulation in grapevine and Arabidopsis. The effects of the electric current between electrodes inserted in the trunk or inflorescence were observed in grapevine and Arabidopsis, respectively. The expression of pathogen-related or pathogen-induced genes, which are involved in SA- and JA-dependent defense pathway, were analyzed in the leaves. In the grapevine, the disease infection on bunches and leaves was also observed. The results showed that the electric current stimulation affected the SA-dependent defense pathway and increased the defense to the pathogens. It is an interesting experimental challenge on grapevine cultivation. However, in the current version of the manuscript, I think the method explanation and result information are inadequate. The manuscript should be revised. I have questions and suggestions as follows;

  1. In Materials and Methods, more detailed explanations about electrical stimulation treatment are preferred. How was the thickness (inserted size) of electrodes for grapevine and Arabidopsis respectively? I feel the relative size of electrode insertion to the plant (trunk or inflorescence) tissues may affect the intensity of the stimulation or wounding stress. If possible, some photo images and delineations of the electrode insertion parts of grapevine and Arabidopsis should be shown.

  1. How much electricity was totally applied for the stimulation treatment? I think there were fluctuations in the electrical voltage supplied from the solar panel. These fluctuations may also be kind of a stimulation. Is there any data about monitoring the electric quantity supplied from the solar panel in each experiment?

  1. How is the static electrification in each part of the experimental system? The environmental condition is thought to significantly affect the level (intensity or quantity) of the stimulation. The soil composition, weather, humidity, charged particles around the plant, and watering, etc. many factors may be involved in the physiological responses related to the electric stimulation in this experiment. To clarify the electric stimulation that had occurred, explanations about those other factors affecting the plant response should be done.

  1. Figure 4 indicates an over discussion from the presented results. Any experimental data about signaling pathways of SA and JA in the grapevine are not shown in this manuscript. It is too aggressive discussions integrating the results from different plant species. If the authors want to discuss the two species together, the same genes should be analyzed and confirmed in grapevine and Arabidopsis.

[Minor revisions]

In Figure 2 and Figure 3, the gene expression data are indicated by “Relative intensity”. Other words, such as “Relative gene expression” or “Relative expression level”, are probably better.

In Figure 3, the scales of the y-axis in panel B and panel C (gene expression data) are shown on a logarithmic scale. It is different from the similar charts in Figure 2 panel B. Why are those representation methods different? Especially, the bar chart of Figure 3 panel C looks strange because a logarithmic scale is used.

Author Response

Point-by-point response to the comments (Reviewer #3):

Reviewers' comments:

  1. This article reports about the responses to the electrical stimulation in grapevine and Arabidopsis. The effects of the electric current between electrodes inserted in the trunk or inflorescence were observed in grapevine and Arabidopsis, respectively. The expression of pathogen-related or pathogen-induced genes, which are involved in SA- and JA-dependent defense pathway, were analyzed in the leaves. In the grapevine, the disease infection on bunches and leaves was also observed. The results showed that the electric current stimulation affected the SA-dependent defense pathway and increased the defense to the pathogens. It is an interesting experimental challenge on grapevine cultivation. However, in the current version of the manuscript, I think the method explanation and result information are inadequate. The manuscript should be revised. I have questions and suggestions as follows;

Answer: Thank you very much for your advice to improve our manuscript. The manuscript was revised according to the reviewers’ suggestions. We hope that our revision is satisfactory and the revised manuscript is now acceptable for publication in the Plants.

  1. In Materials and Methods, more detailed explanations about electrical stimulation treatment are preferred. How was the thickness (inserted size) of electrodes for grapevine and Arabidopsis respectively? I feel the relative size of electrode insertion to the plant (trunk or inflorescence) tissues may affect the intensity of the stimulation or wounding stress. If possible, some photo images and delineations of the electrode insertion parts of grapevine and Arabidopsis should be shown.

Answer: We have taken your suggestion, and revised the Materials and methods section in the revised manuscript. On the other hand, we could NOT give any photo images and of delineations electrode insertion, because electrode was entirely screwed or pricked into the trunks of grapevines and seedlings and the inflorescence of Arabidopsis plants. Information of each electrodes (length and diameter) was included in each sub-section.

(See the Materials and methods section, please)

  1. How much electricity was totally applied for the stimulation treatment? I think there were fluctuations in the electrical voltage supplied from the solar panel. These fluctuations may also be kind of a stimulation. Is there any data about monitoring the electric quantity supplied from the solar panel in each experiment?

Answer: We investigated the oscillation of voltage applied to a field-grown grapevine by electrical stimulation (Mikami et al. 2017. Scientia Horticulturae 226:285–292, reference #11). Voltage from the solar panels connected to a grapevine through the electrodes was correlated with illuminance on the panels from solar radiation (the Figure below). Illuminance exceeding 8,000 lux induced full capacity of the two solar panels (approximately 10 V), whereas voltage was low when illuminance was below 8,000 lux. No voltage was detected on grapevines that had only electrodes without the solar panel. Th performances of electrical stimulation on grapevine seedling and Arabidopsis plant were similar to field-grown grapevines.

In the Results section, we added the description as follows:

‘In our system, illuminance exceeding 8,000 lux induced full capacity of a solar panel, whereas voltage was low when illuminance was below 8,000 lux and no voltage was detected on grapevines that had only electrodes without solar panel [11].’

(See p. 2, lines 70-73, please)

  1. How is the static electrification in each part of the experimental system? The environmental condition is thought to significantly affect the level (intensity or quantity) of the stimulation. The soil composition, weather, humidity, charged particles around the plant, and watering, etc. many factors may be involved in the physiological responses related to the electric stimulation in this experiment. To clarify the electric stimulation that had occurred, explanations about those other factors affecting the plant response should be done.

Answer: Oscillation of voltages applied to a field-grown grapevine by electrical stimulation was shown in comment #3. However, as you commented, one important question that remains to be clarified is whether other environmental factors including grapevine cultivars affect the physiological responses related to the electric stimulation. Sine we can’t answer this question at present, we will perform next experiments in a number of vineyards having different environmental conditions and then analyze the correlation between each environmental factor and the efficacy by electrical stimulation.

In the Discussion section, we added the description as follows:

‘Voltage applied to a field-grown grapevine by electrical stimulation was oscillated by illuminance [11]. One important question that remains to be clarified is whether other environmental factors including soil composition, weather, training system, and grapevine cultivars affect the physiological responses related to the electric stimulation. To explore further the applicability of electrical stimulation to disease control in viticulture, field tests on a number of vineyards having different environmental conditions and cultivars, adjustments of starting time and conditions for electrical stimulation, and development of a universal electrical stimulation apparatus are required.’

(See p. 9, lines 238-246, please)

  1. Figure 4 indicates an over discussion from the presented results. Any experimental data about signaling pathways of SA and JA in the grapevine are not shown in this manuscript. It is too aggressive discussions integrating the results from different plant species. If the authors want to discuss the two species together, the same genes should be analyzed and confirmed in grapevine and Arabidopsis.

Answer: We agree with you about Figure 4. We could not show any results about SA generation and transport in grapevine by our technical issues. We added the description about it as follows:

‘Further investigations are necessary to elucidate whether SA generates at tissues exposed to electrical stimulation and transports into phloem of grapevine as well as how grapevine cells recognize electrical current and how the recognition activates SA biosynthesis.’

(See p. 9, lines 217-220, please)

 Also, we modified Figure 4 for readers not to jump to conclusions.

       (See the revised Figure 4, please)

  1. In Figure 2 and Figure 3, the gene expression data are indicated by “Relative intensity”. Other words, such as “Relative gene expression” or “Relative expression level”, are probably better.

Answer: We have taken your suggestion, and changed ‘Relative intensity’ to ‘Relative gene expression’ in the revised Figures 2 and 3.

       (See revised Figure 2 and Figure 3, please)

  1. In Figure 3, the scales of the y-axis in panel B and panel C (gene expression data) are shown on a logarithmic scale. It is different from the similar charts in Figure 2 panel B. Why are those representation methods different? Especially, the bar chart of Figure 3 panel C looks strange because a logarithmic scale is used.

Answer: Arabidopsis plants induced more transcripts (relative values to actin.) of plant defense-related genes in response to electrical stimulation than grapevine seedlings. Since normal scales in Y-axes hid slight changes of gene expression, we used logarithmic scale in Y-axes in Figures 3B and 3C.

Round 2

Reviewer 3 Report

I checked the revised points in the manuscript and understood the authors’ reply comment. However, I can’t agree with the reply about the scale setting for Figures 3B and 3C.

Although the logarithmic scale is used in Y-axes because the normal scales hid slight changes in gene expression, Figure 3C has a problem of misunderstanding the data and looks like a cheating display method by the logarithmic scale. The originations of boxed data bars should be “1.0”. Because all values are less than 1.0, the data bars are in downward directions if the logarithmic scale is used. For the current version of Figure 3C, the design and displaying method look very strange.

I have a further question about the calculation method of the relative gene expression. How were the relative values calculated? Which sample was set to “1.0” as a standard? In Figure 2B, I understand the “Control - 0 day” is set to “1.0”. Whereas, in Figures 3B and 3C, there is no indication and explanation about the standard sample set as “1.0”. The data display method of Figures 3B and 3C should be reconsidered.

Figure 4 is an illustration based on the authors’ predictions (speculations). This illustration makes for over discussion and misleading of the interpretation. I suggest Figure 4 should be removed from this manuscript, if the authors will not revise it. The authors’ predictions and ideas are already described enough in the main text.

Author Response

Point-by-point response to the comments (Reviewer #3):

Reviewers' comments:

  1. I checked the revised points in the manuscript and understood the authors’ reply comment. However, I can’t agree with the reply about the scale setting for Figures 3B and 3C.

Although the logarithmic scale is used in Y-axes because the normal scales hid slight changes in gene expression, Figure 3C has a problem of misunderstanding the data and looks like a cheating display method by the logarithmic scale. The originations of boxed data bars should be “1.0”. Because all values are less than 1.0, the data bars are in downward directions if the logarithmic scale is used. For the current version of Figure 3C, the design and displaying method look very strange.

Answer:  Thank you very much for the reviewer’s advice to improve our manuscript. The manuscript was revised according to the reviewers’ suggestions. We hope that our revision is satisfactory and the revised manuscript is now acceptable for publication in the Plants.

                 The calculation method for relative gene expression of each gene was explained in comment #2 below. In regard to Figure 3C, we agree with the reviewer’s suggestion and revised Y-axes in Figure 3C from logarithmic scale to normal scale.

             (See the revised Figure 3C, please)

  1. I have a further question about the calculation method of the relative gene expression. How were the relative values calculated? Which sample was set to “1.0” as a standard? In Figure 2B, I understand the “Control - 0 day” is set to “1.0”. Whereas, in Figures 3B and 3C, there is no indication and explanation about the standard sample set as “1.0”. The data display method of Figures 3B and 3C should be reconsidered.

Answer:  In this manuscript (Figures 2B, 3B and 3C), the relative gene expression was calculated by same methods. We used each actin expression value in each sample at each sampling time (0, 10, and 20 days in Figure 2B, 0, 12, 24 and 48 h in Figure 3B) as a standard (“1.0”) and calculated the expression values of each gene in each sample at each sampling time. We added the information in the Materials and methods as follows:

         ‘Relative gene expression was expressed as relative values to actin expression values at each sampling time.’

         (See p. 11, lines 328-329, please)

            In addition, we replied to your suggestion related to Figure 3C in comment #1 above.

  1. Figure 4 is an illustration based on the authors’ predictions (speculations). This illustration makes for over discussion and misleading of the interpretation. I suggest Figure 4 should be removed from this manuscript, if the authors will not revise it. The authors’ predictions and ideas are already described enough in the main text.

Answer:  We have taken reviewer’s suggestion and removed Figure 4 from the revised manuscript.

Round 3

Reviewer 3 Report

I read the revised manuscript and the authors’ reply. I think this manuscript is acceptable in the current version.